# Sophocarpine Alleviates Isoproterenol-Induced Kidney Injury by Suppressing Inflammation, Apoptosis, Oxidative Stress and Fibrosis

**DOI:** 10.3390/molecules27227868

**Published:** 2022-11-15

**Authors:** Wei Zhou, Yang Fu, Jin-Song Xu

**Affiliations:** 1Department of Cardiovascular Medicine, The Second Affiliated Hospital of Nanchang University, No. 1 Minde Road, Nanchang 330006, China; 2Key Laboratory of Molecular Biology in Jiangxi Province, Nanchang 330006, China

**Keywords:** sophocarpine, inflammation, apoptosis, oxidative stress, fibrosis

## Abstract

One of the most common diseases affecting people and leading to high morbidity is kidney injury. The alleviation of inflammation and apoptosis is considered a potential therapeutic approach for kidney injury. Sophocarpine (SOP), a tetracyclic quinolizidine alkaloid, exhibits various beneficial biological properties. To investigate the effects of SOP on isoproterenol (ISO)-induced kidney injury, we randomly divided mice into four groups: Control, ISO, ISO+SOP (20 mg/kg) and ISO+SOP (40 mg/kg). SOP was administered intraperitoneally to the mice over two weeks, accompanied by intraperitoneal stimulation of ISO (10 mg/kg) for another four weeks. After the mice were sacrificed, several methods such as ELISA, staining (H&E, TUNEL, DHE and Masson) and Western blotting were applied to detect the corresponding indicators. The kidney injury serum biomarkers SCr and BUN increased after the ISO challenge, while this effect was reversed by treatment with SOP. Pathological changes induced by ISO were also reversed by treatment with SOP in the staining. The inflammatory cytokines IL-β, IL-6, TNF-α, MCP-1 and NLRP3 increased after the challenge with ISO, while they were decreased by treatment with SOP. The apoptotic proteins cleaved-caspase-3 and Bax increased, while Bcl-2 decreased, after the challenge with ISO, and these effects were reversed by treatment with SOP. The antioxidant proteins SOD-1 and SOD-2 decreased after being stimulated by ISO, while they increased after the treatment with SOP. The fibrotic proteins collagen I, collagen III, α-SMA, fibronectin, MMP-2 and MMP-9 increased after the challenge with ISO, while they decreased after the treatment with SOP. We further discovered that the TLR-4/NF-κB and TGF-β1/Smad3 signaling pathways were suppressed, while the Nrf2/HO-1 signaling pathway was activated. In summary, SOP could alleviate ISO-induced kidney injury by inhibiting inflammation, apoptosis, oxidative stress and fibrosis. The molecular mechanisms were suppression of the TLR-4/NF-κB and TGF-β1/Smad3 signaling pathways and activation of the Nrf2/HO-1 signaling pathway, indicating that SOP might serve as a novel therapeutic strategy for kidney injury.

## 1. Introduction

The deterioration of renal function characterizes kidney injury, which is often associated with a significant increase in serum creatinine (SCr) and blood urea nitrogen (BUN) levels, as well as a persistent decrease in the glomerular filtration rate (GFR) [1]. The course of this disease is temporary, but in the long term, kidney dysfunction is irreversible [2]. There has been a progressive increase in morbidity and mortality due to kidney injury, with nearly 2 million deaths occurring every year worldwide, contributing to a serious medical security issue [3]. It has been reported that patients with kidney injury may eventually develop chronic kidney disease (CKD) [4]. Although the pathological changes in kidney injury have been extensively studied over the last few decades, few of the permitted kidney-injury-specific therapeutic approaches are available at present. Therefore, it is urgent to identify and develop novel and effective therapeutic approaches for kidney injury.

Sophocarpine, extracted from *Sophora flavescens*, is a tetracyclic quinolizidine alkaloid that exhibits various beneficial biological properties [5]. Research has found that sophocarpine exhibits anti-inflammatory, anti-tumor, anti-nociceptive, neuroprotective and immune regulatory properties [6,7,8,9]. It has also been reported that sophocarpine could attenuate lipopolysaccharide-induced liver injury primarily by suppressing oxidative stress, inflammation and apoptosis [10]. A further study reported that sophocarpine exerted a protective effect on LPS-induced acute lung injury [11]. Moreover, Li et al. discovered that sophocarpine also alleviated murine lupus nephritis by suppressing the NLRP3 inflammasome and NF-κB activation [12]. Isoproterenol (ISO), which acts as a nonselective beta adrenoceptor agonist, is well-known for inducing cardiac dysfunction and some related renal diseases [13]. Surprisingly, detailed studies on the effects of sophocarpine on the occurrence and progression of isoproterenol (ISO)-induced kidney injury have not been conducted. Thus, in the present study, we used ISO to construct a kidney injury model and further studied the effects of sophocarpine on the occurrence and advancement of ISO-induced kidney injury.

## 2. Results

### 2.1. Treatment with Sophocarpine (SOP)-Alleviated Isoproterenol (ISO)-Induced Kidney Injury

All of the treatments applied to the mice in this study are summarized in Figure 1A. The levels of SCr and BUN in the serum were detected after the mice were sacrificed. The serum levels of SCr and BUN in the ISO group were higher than those recorded in the Control group, while we found a significant decrease in the serum levels of these two indices in the ISO+Sophocarpine (20 mg/kg) and ISO+Sophocarpine (40 mg/kg) groups following the treatment with SOP (Figure 2B,C). These results indicate that treatment with SOP could alleviate ISO-induced kidney injury in mice. Moreover, we also detected the serum levels of the cardiac injury biomarkers CK-MB and LDH and found that treatment with SOP could also alleviate ISO-induced cardiac injury in mice (Figure 2D,E).

### 2.2. Treatment with SOP Ameliorated ISO-Induced Inflammatory Response

An inappropriate inflammatory response contributes to pathological alterations in kidney tissues challenged by ISO. The results of the H&E staining assay show that the extent of ISO-based damage in the renal structures was higher in the ISO group than in the Control group. Moreover, the kidney tissues in the ISO group also showed more severe cellular edema than in the Control group. Meanwhile, these effects were significantly reversed by treatment with SOP (Figure 2A). We further used an enzyme-linked immunosorbent assay (ELISA) and Western blotting assay to detect the levels of inflammatory cytokines. The serum levels of IL-β, IL-6 and TNF-α increased significantly after the challenge with ISO in the ISO group, while these effects were significantly inhibited by treatment with SOP in the ISO+Sophocarpine (20 mg/kg) and ISO + Sophocarpine (40 mg/kg) groups (Figure 2B–D). It was also observed that the protein expression levels of IL-β, IL-6, TNF-α, MCP-1 and NLRP3 increased in the kidney tissues after the challenge with ISO in the ISO group compared to the Control group. Treatment with SOP significantly reduced the corresponding indices in the ISO+Sophocarpine (20 mg/kg) and ISO + Sophocarpine (40 mg/kg) groups (Figure 2E–J). As previously recorded, the TLR-4/NF-κB signaling pathway could regulate inflammatory responses. Therefore, we studied the effects of SOP administration on the activation of the TLR-4/NF-κB signaling pathway. It was observed that treatment with SOP significantly hindered the ISO-induced increase in TLR-4 and phosphorylation of NF-κB (Figure 2E,K,L). These results suggest that inhibition of the activation of the TLR-4/NF-κB signaling pathway, which was attributed to the treatment with SOP, could address the problem of ISO-induced renal inflammation in mice.

### 2.3. Treatment with SOP Hindered ISO-Induced Apoptosis

A TUNEL staining assay and Western blotting assay were used in this study to comprehend the effects of the treatment with SOP on ISO-induced renal apoptosis. The results obtained from the TUNEL staining assay reveal that the level of apoptosis in the ISO group was significantly higher than that in the Control group. Meanwhile, the level of apoptosis was significantly decreased in the ISO+Sophocarpine (20 mg/kg) and ISO+Sophocarpine (40 mg/kg) groups (Figure 3A,B). Apoptosis-related proteins (cleaved-caspase 3, Bax and Bcl-2) were studied using a Western blotting assay. It was observed that the protein levels of cleaved-caspase 3 and Bax recorded for the ISO group were significantly higher compared to the those in the Control group. Significant suppression of these effects was observed in the ISO+Sophocarpine (20 mg/kg) and ISO+Sophocarpine (40 mg/kg) groups under the conditions of the treatment with SOP (Figure 3C–E). The protein level of Bcl-2 in the Control group was higher than the protein level of Bcl-2 in the ISO group. The protein level of Bcl-2 was highly promoted in the ISO+Sophocarpine (20 mg/kg) and ISO+Sophocarpine (40 mg/kg) groups (Figure 3C,F). These results demonstrate that treatment with SOP attenuated ISO-induced renal apoptosis in mice.

### 2.4. Treatment with SOP Inhibited ISO-Induced Oxidative Stress

Oxidative stress exhibits an important influence on the progression of ISO-induced kidney injury. A DHE staining assay was conducted to study whether treatment with SOP affected the level of ISO-induced oxidative stress. Compared to the Control group, the total superoxide level detected with the DHE staining assay was apparently increased in the ISO group, while it was suppressed by treatment with SOP in the ISO + Sophocarpine (20 mg/kg) and ISO+Sophocarpine (40 mg/kg) groups (Figure 4A,B). The concentrations of MDA, SOD and GSH in the serum were used to determine the redox status. Compared to the Control group, the ISO group exhibited a higher level of MDA and lower levels of SOD and GSH, while these effects were suppressed by treatment with SOP (Figure 4C–E). As regards the antioxidant protein levels of SOD-1 and SOD-2, the results of the Western blotting assay show that ISO apparently reduced these antioxidant protein levels in the ISO group compared to the Control group, while treatment with SOP significantly increased these antioxidant protein levels in the ISO+Sophocarpine (20 mg/kg) and ISO+Sophocarpine (40 mg/kg) groups (Figure 4F–H). The Nrf2/HO-1 signaling pathway exhibits an important role in the progression of oxidative stress. Thus, we studied the effects of treatment with SOP on the activation of the Nrf2/HO-1 signaling pathway and found that treatment with SOP significantly reversed the ISO-induced reduction in Nrf2 and HO-1 (Figure 4F,I,J). In summary, treatment with SOP might protect against ISO-induced kidney injury by suppressing renal oxidative stress by activating the Nrf2/HO-1 signaling pathway in mice.

### 2.5. Treatment with SOP Reduced ISO-Induced Fibrosis

Renal fibrosis displays major pathological alterations in kidney injury induced by ISO. Therefore, we studied the influences of treatment with SOP on renal fibrosis induced by ISO. First, the results of the Masson staining assay reveal that the area of fibrosis in the kidney tissues increased significantly when challenged by ISO in the ISO group compared to the Control group, while treatment with SOP apparently reversed this effect in the ISO+Sophocarpine (20 mg/kg) and ISO+Sophocarpine (40 mg/kg) groups (Figure 5A). Second, we used a Western blotting assay to measure the fibrosis-related protein levels. The protein levels of collagen I, collagen III, α-SMA, fibronectin, MMP-2 and MMP-9 increased significantly when challenged by ISO compared to those in the Control group, while these effects were inhibited by treatment with SOP (Figure 5B,C,E–J). Furthermore, the TGF-β1/Smad3 signaling pathway, a classical signaling pathway of fibrosis, was investigated in the present study. The results of the Western blotting assay show that treatment with SOP suppressed the increase in TGF-β1 and phosphorylation of Smad3 induced by ISO (Figure 5D,K,L). These results suggest that treatment with SOP could suppress ISO-induced renal fibrosis by inhibiting the TGF-β1/Smad3 signaling pathway in mice.

## 3. Discussion

Kidney injury is a complex disease that is characterized by the impairment of renal function. It has been previously reported that a variety of pathogenic factors can lead to kidney injury, such as some drugs and a majority of toxics [14]. Moreover, sepsis has also been reported to serve as an important pathogenic factor that contributes to kidney injury [15]. Isoproterenol (ISO), which is a nonselective beta adrenoceptor agonist, can be used to induce various cardiac dysfunctions such as myocardial infarction, cardiac hypertrophy or heart failure [16,17,18]. It has also been reported that ISO can cause kidney injury [19]. Furthermore, growing evidence suggests that patients suffering from kidney injury might develop chronic kidney disease (CKD) [4]. Due to the fact that few of the permitted kidney-injury-specific therapeutic approaches are available at present, it is urgent for us to identify and develop novel and effective therapeutic approaches for kidney injury. Sophocarpine (SOP) is a tetracyclic quinolizidine alkaloid extracted from *Sophora flavescens,* and shows a variety of biological characteristics such as anti-inflammatory, anti-tumor, anti-nociceptive, neuroprotective and immune regulatory properties [5,6,7,8,9]. It has been reported that sophocarpine could attenuate liver injury by suppressing oxidative stress, inflammation and apoptosis [10]. Another study reported that sophocarpine exerted a protective effect on acute lung injury [11]. Moreover, Li et al. discovered that sophocarpine also alleviated murine lupus nephritis by suppressing the NLRP3 inflammasome and NF-κB activation [12]. Betaine, also a natural product, was found to have a protective impact on renal function in ISO-induced myocardial infarction (MI) [20]. In the present study, we studied the effects of treatment with SOP on ISO-induced kidney injury and further investigated the underlying molecular mechanisms, finding that SOP could alleviate ISO-induced kidney injury by inhibiting inflammation, apoptosis, oxidative stress and fibrosis (Figure 6).

The serum levels of SCr and BUN are characteristic biomarkers of kidney injury, which will apparently increase when kidney injury occurs [21]. In the present study, we found that the serum levels of SCr and BUN apparently increased when the mice were injected with ISO, while treatment with SOP reversed these effects induced by ISO and prevented the deterioration of kidney injury. The results suggest that treatment with SOP alleviated ISO-induced kidney injury in mice. We also found that treatment with SOP could alleviate ISO-induced cardiac injury, as evidenced by the downregulation of serum levels of the cardiac injury biomarkers CK-MB and LDH when the mice were treated with SOP.

An inappropriate inflammatory response contributes to pathological alterations in kidney tissues [22]. The results of the present study show that injection with ISO led to renal structural destruction and cellular edema as well as an increase in the serum levels of inflammatory cytokines such as IL-β, IL-6 and TNF-α, while these effects were significantly inhibited by treatment with SOP. We further detected the protein expression levels of some inflammatory cytokines such as IL-β, IL-6, TNF-α, MCP-1 and NLRP3 and discovered that treatment with SOP significantly reduced the corresponding elevated indicators induced by ISO. As indicated in previous research, the TLR-4/NF-κB signaling pathway could regulate inflammatory responses [23]. It was observed that treatment with SOP significantly hindered the ISO-induced increase in TLR-4 and phosphorylation of NF-κB. These results indicate that treatment with SOP could suppress ISO-induced renal inflammation by inhibiting the TLR-4/NF-κB signaling pathway.

It has also been reported that apoptosis influences the pathological changes associated with kidney injury [24]. In the present study, we used a TUNEL staining assay and Western blotting assay to comprehend the effects of treatment with SOP on ISO-induced renal apoptosis. The TUNEL staining revealed that ISO apparently increased the level of apoptosis, while it was significantly inhibited by treatment with SOP. The expression levels of apoptosis-related proteins (cleaved-caspase 3, Bax and Bcl-2) were detected in this study. We discovered that the protein levels of cleaved-caspase 3 and Bax significantly increased when the mice were injected with ISO, while treatment with SOP markedly suppressed these effects. The protein expression trend of Bcl-2 was contrary to that of cleaved-caspase 3 and Bax. These results reveal that treatment with SOP could attenuate ISO-induced renal apoptosis.

Oxidative stress also exhibits an important influence on the progression of kidney injury [25]. The DHE staining conducted in this study showed that treatment with SOP reversed the upregulation of the total superoxide level stimulated by ISO. Moreover, administration of ISO contributed to a higher level of MDA and lower levels of SOD and GSH, which are well-known for serving as serum biomarkers of oxidative stress [26], while these effects were suppressed by treatment with SOP. As regards the antioxidant protein expression levels of SOD-1 and SOD-2, we found that ISO apparently reduced these antioxidant protein expression levels, while treatment with SOP significantly increased them. The Nrf2/HO-1 signaling pathway exhibits an important role in the progression of oxidative stress [27]. Thus, we investigated the influences of treatment with SOP on the activation of the Nrf2/HO-1 signaling pathway and discovered that treatment with SOP apparently inhibited the ISO-induced reduction in Nrf2 and HO-1. In sum, treatment with SOP might protect against ISO-induced renal oxidative stress by activating the Nrf2/HO-1 signaling pathway.

Fibrosis is also a major pathological alteration in kidney injury [28]. In the present study, Masson staining revealed that the area of fibrosis in the kidney tissues increased significantly when challenged by ISO, while treatment with SOP apparently reversed this effect. Second, we further measured fibrosis-related protein expression levels, such as collagen I, collagen III, α-SMA, fibronectin, MMP-2 and MMP-9, and discovered that these protein expression levels increased significantly when stimulated by ISO, while these effects were inhibited by treatment with SOP. Furthermore, the TGF-β1/Smad3 signaling pathway, a classical signaling pathway of fibrosis [29], was also investigated in the present study. The results show that treatment with SOP suppressed the increase in TGF-β1 and phosphorylation of Smad3 induced by ISO. These results suggest that treatment with SOP could suppress ISO-induced renal fibrosis by inhibiting the TGF-β1/Smad3 signaling pathway.

Additionally, some limitations exist in the present study. We only conducted an in vivo experiment to investigate the effects of treatment with sophocarpine on ISO-induced kidney injury. Hence, the results shown herein should be verified by conducting in vitro experiments. Moreover, the toxicity of SOP in mice was not evaluated in the present study. Simultaneously, the molecular mechanisms should be investigated deeply to effectively explain the pathophysiological processes involved in this study.

## 4. Materials and Methods

### 4.1. Chemicals and Animals

We used 32 male C57BL/6 mice (weight: 22 ± 3 g; age: 6–8 weeks) that were purchased from the Hunan SJA Laboratory Animal Co., Ltd. (Changsha, China) to conduct the present research. Sophocarpine (SOP, purity: ≥98%) was obtained from MedChemExpress (cat no. 6483-15-4). Isoproterenol (ISO) was obtained from Shanghai Harvest Pharmaceutical Co., Ltd. (Shanghai, China) (cat no. 7683-59-2). All other reagents used in this study were commercially available.

### 4.2. Animal Model

All mice in this study were reared at the Animal Center of the Jiangxi Medical College (Nanchang University). The mice were subjected to conditions of a 12 h/12 h light/dark cycle, with free access to water and food. The temperature and humidity of the feeding environment were maintained at 22 ± 3 °C and 54 ± 6%, respectively. At the start of the experiment, the mice were adjusted to the conditions of the raising environment over a period of approximately 7 days. However, at this stage, they were not subjected to any experimental treatment conditions. The mice under this study were randomly divided into four groups. Control (n = 8): phosphate-buffered saline (PBS) (devoid of other reagents) was intraperitoneally (i.p.) administered to the mice for four weeks; ISO (n = 8): the mice belonging to this group were injected i.p. with ISO (10 mg/kg) once per day for four weeks; ISO+SOP (20 mg/kg) (n = 8): ISO (10 mg/kg) and SOP (20 mg/kg) were injected i.p. into the mice belonging to this group once per day over two weeks, followed by intraperitoneal stimulation of ISO (10 mg/kg) for another two weeks; ISO+SOP (40 mg/kg) (n = 8): ISO (10 mg/kg) and SOP (40 mg/kg) were injected i.p. into the mice belonging to this group once per day over two weeks, followed by intraperitoneal stimulation of ISO (10 mg/kg) for another two weeks. The dosage and procedure of the reagent administration were carried out according to previous research [6,30]. Approval for the treatment protocols was obtained from the Animal Care and Use Committee of the Second Affiliated Hospital of Nanchang University (China). We diligently followed the relevant guidelines and regulations to conduct the experiments.

### 4.3. Detection of Serum SCr, BUN, CK-MB and LDH Levels

Whole blood was collected following the completion of the treatment procedures. After anesthetization by pentobarbital sodium (30 mg/kg) intraperitoneally, vacuum tubes were used to collect blood from the right ventricle of the mice, and then the mice were sacrificed for further study. The serum was extracted from the whole blood following the process of centrifugation (4000× *g*; 30 min; 4 °C). A BS-800 Chemistry Analyzer (Mindray, Shenzhen, China) was used to detect SCr and BUN levels, and an automatic biochemical analyzer (Chemray240, Rayto, Shenzhen, China) was used to investigate serum levels of cardiac injury biomarkers, namely, CK-MB and LDH. Moreover, the kidney tissues were extracted from the mice for the purpose of conducting further study.

### 4.4. Hematoxylin/Eosin (H&E) and Masson Staining

Kidney tissues were isolated from the mice immediately after they were sacrificed. The left kidney tissues were fixed immediately with 4% formaldehyde. Then, the kidney tissues were dehydrated with alcohol and embedded in paraffin (5 μm), followed by the processes of hematoxylin/eosin (H&E) and Masson staining. An optical microscope was used to observe the pathological changes in kidney tissues.

### 4.5. Measurement of Inflammatory Cytokines by Relevant Enzyme-Linked Immunosorbent Assay (ELISA) Kits

The serum levels of the inflammatory cytokines IL-β, IL-6 and TNF-α were detected using specific mouse ELISA kits. The detailed information of these specific mouse ELISA kits is as follows: IL-1β (cat no. 88-7013; Invitrogen, CA, USA), IL-6 (cat no. 88-7064; Invitrogen, CA, USA) and TNF-α (cat no. 88-7324; Invitrogen, CA, USA). The experimental process strictly followed the instructions outlined by the manufacturer.

### 4.6. Terminal Deoxynucleotidyl Transferase dUTP Nick End Labeling (TUNEL) Staining

As described above, procedures regarding the paraffin sections of the kidney tissues were carried out, and these sections were then deparaffinated using toluene. We used different gradients of ethanol to dehydrate the samples following standard procedures. The TUNEL staining assay was conducted using a fluorescent detection kit (cat no. 11684817910, Roche, Shanghai, China). The procedures strictly followed the instructions provided by the manufacturer.

### 4.7. Dihydroethidium (DHE) Staining

The conditions of renal reactive oxygen species (ROS) of different groups were detected using a DHE staining assay. First, we used DHE solution to treat the frozen slices of kidney tissues and then incubated these slices at 37 °C for about 30 min in a dark environment. Then, we applied DAPI (1 mg/mL) to counterstain the cell nucleus of the kidney tissues for 5 min. Following this, these slices were washed with water and measured using a laser scanning confocal microscope (Olympus FV1200, Tokyo, Japan).

### 4.8. Detection of Serum Malondialdehyde (MDA), Superoxide Dismutase (SOD) and Glutathione (GSH) Levels

The serum extracted from the whole blood was used to evaluate the levels of oxidative stress of the mice in these four groups. The experimental process strictly followed the instructions outlined by the manufacturer, and the absorbance of the wells was measured at 532 nm (MDA), 560 nm (SOD) and 412 nm (GSH) using a microplate reader (Thermo Fisher Scientific). The detailed information of these specific mouse kits is as follows: MDA (cat no. A003-1; Nanjing Jiancheng Bioengineering Institute, Nanjing, China), SOD (cat no. A001-3; Nanjing Jiancheng Bioengineering Institute, Nanjing, China) and GSH (cat no. A006-2-1; Nanjing Jiancheng Bioengineering Institute, Nanjing, China).

### 4.9. Western Blotting

The total protein was extracted from the kidney tissues using a protein extraction kit (cat no. P0013B; Beyotime Biotechnology, Jiangsu, China). A bicinchoninic acid protein assay kit (cat no. P0012; Beyotime Biotechnology, Jiangsu, China) was applied to detect the concentrations of the total protein. Then, the total protein was divided into certain proteins using the sodium dodecyl sulfate-polyacrylamide gel electrophoresis (SDS-PAGE) method. Subsequently, these proteins were transferred onto polyvinylidene fluoride (PVDF) membranes (EMD Millipore, Billerica, MA, USA). Primary antibodies were incubated with the PVDF membranes, which contained certain target proteins (incubation time: 24 h; temperature: 4 °C). Tris-buffered saline (TBST; supplemented with 0.1% Tween-20) was used to wash the PVDF membranes for about 30 min, and then the appropriate secondary antibodies (goat anti-rabbit IgG, cat no. B900210, 1:5000, Proteintech, Rosemont, IL, USA; goat anti-mouse IgG, cat no. 15014, 1:5000, Proteintech, Rosemont, IL, USA) were incubated with the washed membranes (incubation time: 1 h). Finally, certain protein bands were detected using the scanner and the enhanced chemiluminescence detection kit (Thermo Fisher Scientific, Waltham, MA, USA). The detailed information of these primary antibodies is as follows: anti-IL-1β (cat no. ab200478; 1:1000; Abcam, Cambridge, MA, USA), anti-IL-6 (cat no. ab6672; 1:1000; Abcam, Cambridge, MA, USA), anti-TNF-α (cat no. ab1793; 1:1000; Abcam, Cambridge, MA, USA), anti-MCP-1 (cat no. ab7202; 1:1000; Abcam, Cambridge, MA, USA), anti-NLRP3 (cat no. 19771-1-AP; 1:1000; Proteintech, Rosemont, IL, USA), anti-TLR-4 (cat no. 66350-1-Ig; 1:1000; Proteintech, Rosemont, IL, USA), anti-phosphorylated (p)-NF-κB (cat no. CST-3033S; 1:1000; Cell Signaling Technology, Danvers, MA, USA), anti-NF-κB (cat no. CST-8242S; 1:1000; Cell Signaling Technology, Danvers, MA, USA), anti-cleaved-caspase 3 (cat no. CST-9661S; 1:1000; Cell Signaling Technology, Danvers, MA, USA), anti-Bax (cat no. ab32503; 1:1000; Abcam, Cambridge, MA, USA), anti-Bcl-2 (cat no. ab182858; 1:1000; Abcam, Cambridge, MA, USA), anti-SOD-1 (cat no. CST-65778SF; 1:1000; Cell Signaling Technology, Danvers, MA, USA), anti-SOD-2 (cat no. CST-13141S; 1:1000; Cell Signaling Technology, Danvers, MA, USA), anti-Nrf2 (cat no. ab92946; 1:1000; Abcam, Cambridge, MA, USA), anti-HO-1 (cat no. ab52947; 1:1000; Abcam, Cambridge, MA, USA), anti-Collagen Ⅰ (cat no. 14695-1-AP; 1:1000; Proteintech, Rosemont, IL, USA), anti-Collagen Ⅲ (cat no. 22734-1-AP; 1:1000; Proteintech, Rosemont, IL, USA), anti-α-SMA (cat no. 67735-1-Ig; 1:1000; Proteintech, Rosemont, IL, USA), anti-Fibronectin (cat no. ab2413; 1:1000; Abcam, Cambridge, MA, USA), anti-MMP-2 (cat no. 10373-2-AP; 1:1000; Proteintech, Rosemont, IL, USA), anti-MMP-9 (cat no. 10375-2-AP; 1:1000; Proteintech, Rosemont, IL, USA), anti-TGF-β1 (cat no. ab215715; 1:1000; Abcam, Cambridge, MA, USA), anti-phosphorylated (p)-Smad3 (cat no. CST-9520S; 1:1000; Cell Signaling Technology, Danvers, MA, USA), anti-Smad3 (cat no. CST-8685S; 1:1000; Cell Signaling Technology, Danvers, MA, USA), anti-GAPDH (cat no. 60004-1-Ig; 1:1000; Proteintech, Rosemont, IL, USA). A minimum of three independent Western blotting experiments were performed, and the protein bands were analyzed using Image Lab 4.0.1.

### 4.10. Statistical Analyses

The results obtained from this study are presented as means ± standard deviations (S.D). We applied Prism 7.0 (GraphPad Software, San Diego, CA, USA) to analyze the collected data. The method of analysis of variance (ANOVA) was used to compare the results of different groups. A *p*-value of less than 0.05 was considered statistically significant (* denotes *p*-values < 0.05).

## 5. Conclusions

The present study suggested that SOP could alleviate ISO-induced kidney injury by inhibiting inflammation, apoptosis, oxidative stress and fibrosis. The molecular mechanisms were suppression of the TLR-4/NF-κB and TGF-β1/Smad3 signaling pathways and activation of the Nrf2/HO-1 signaling pathway, indicating that SOP might serve as a novel therapeutic strategy for kidney injury.

## Figures and Tables

**Figure 1 molecules-27-07868-f001:**
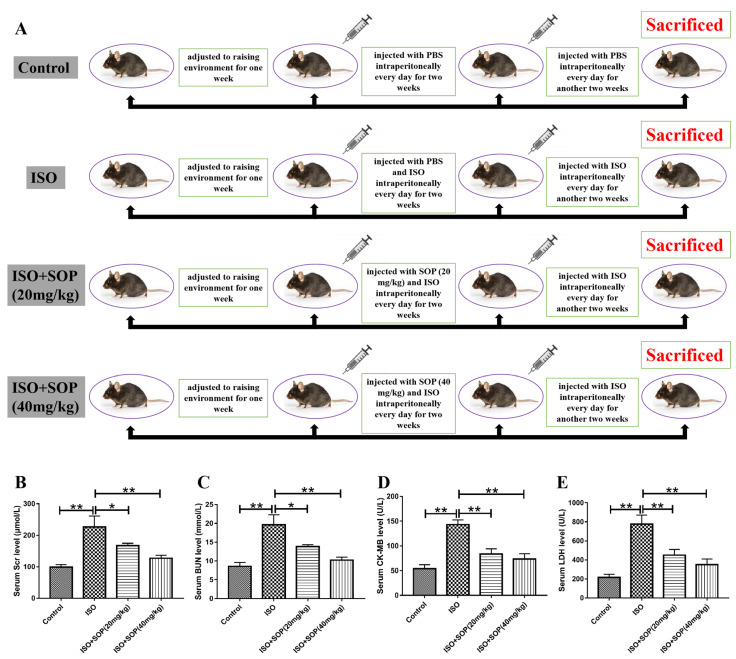
(**A**) Overview of animal treatment protocols applied in the present study. (**B**,**C**) Serum levels of serum creatinine (SCr) and blood urea nitrogen (BUN) were detected (n = 7–8). (**D**,**E**) The cardiac injury biomarkers CK-MB and LDH were also investigated (n = 7–8). * *p* < 0.05 and ** *p* < 0.01. Data represent the mean ± SD of at least three separate experiments.

**Figure 2 molecules-27-07868-f002:**
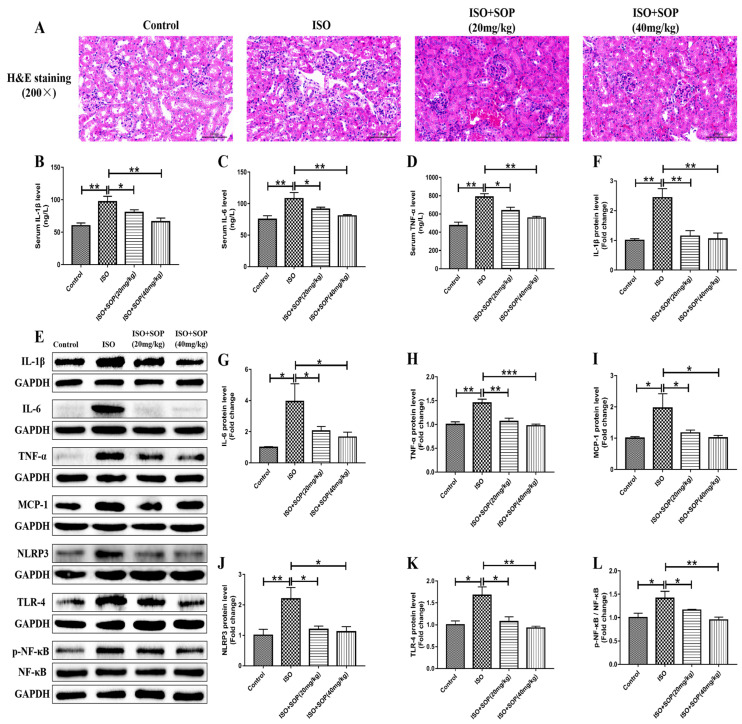
(**A**) An H&E staining assay in kidney tissues was conducted to evaluate pathological changes (n = 7–8). (**B**–**D**) ELISA kits were used to measure serum inflammatory cytokines such as IL-1β, IL-6 and TNF-α (n = 7–8). (**E**–**J**) The protein expression levels of inflammatory cytokines such as IL-1β, IL-6, TNF-α, MCP-1 and NLRP3 were investigated using a Western blotting assay (n = 7–8). (**E**,**K**,**L**) The protein expression levels of TLR-4 and p-NF-κB were also detected using a Western blotting assay (n = 7–8). * *p* < 0.05, ** *p* < 0.01 and *** *p* < 0.001. Data represent the mean ± SD of at least three separate experiments.

**Figure 3 molecules-27-07868-f003:**
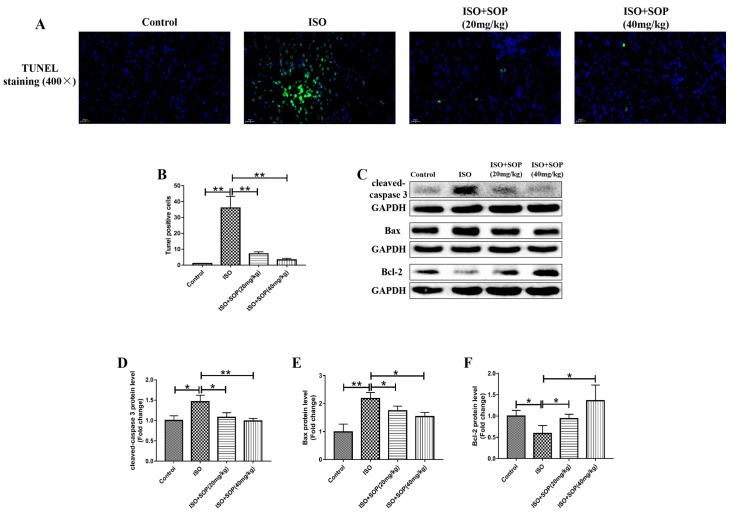
(**A**,**B**) A TUNEL staining assay was conducted to evaluate pathological changes in kidney tissues (n = 7–8). (**C**–**E**) The protein expression levels of apoptosis-related proteins such as cleaved-caspase 3 and Bax were explored using a Western blotting assay (n = 7–8). (**C**,**F**) The protein expression level of the apoptosis-related protein Bcl-2 was also detected using a Western blotting assay (n = 7–8). * *p* < 0.05 and ** *p* < 0.01. Data represent the mean ± SD of at least three separate experiments.

**Figure 4 molecules-27-07868-f004:**
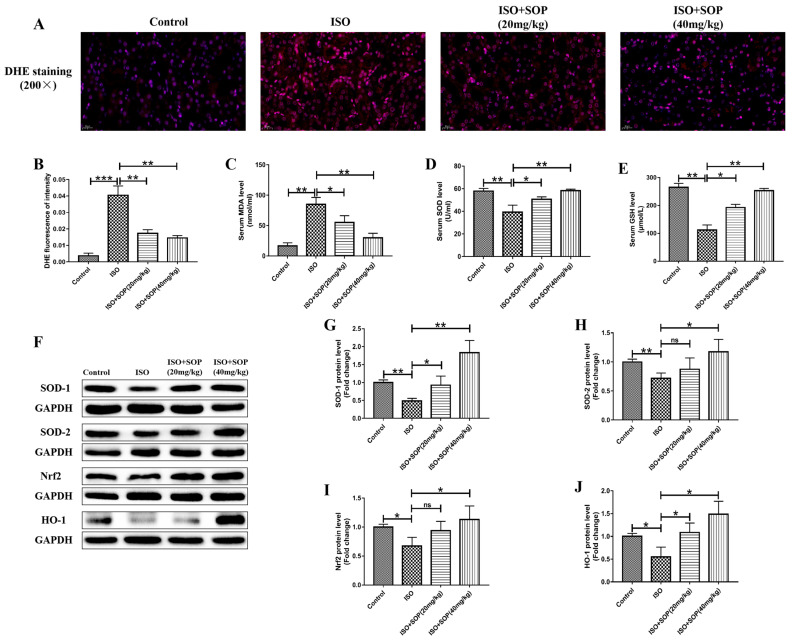
(**A**,**B**) A DHE staining assay was conducted to evaluate pathological changes in kidney tissues (n = 7–8). (**C**–**E**) Certain kits were applied to detect the serum levels of MDA, SOD and GSH (n = 7–8). (**F**–**H**) The protein expression levels of antioxidant proteins such as SOD-1 and SOD-2 were detected using a Western blotting assay (n = 7–8). (**F**,**I**,**J**) The protein expression levels of Nrf2 and HO-1 were also detected using a Western blotting assay (n = 7–8). ns means no statistical significance, * *p* < 0.05, ** *p* < 0.01 and *** *p* < 0.001. Data represent the mean ± SD of at least three separate experiments.

**Figure 5 molecules-27-07868-f005:**
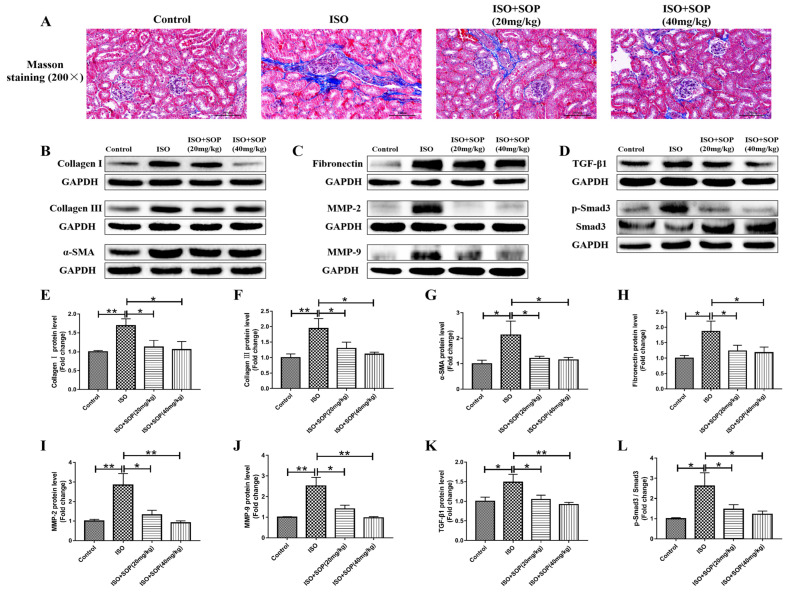
(**A**) A Masson staining assay was used to evaluate the fibrosis area in kidney tissues (n = 7–8). (**B**,**C**,**E**–**J**) The protein expression levels of fibrosis-related proteins such as collagen I, collagen III, α-SMA, fibronectin, MMP-2 and MMP-9 were detected using a Western blotting assay (n = 7–8). (**D**,**K**,**L**) The protein expression levels of TGF-β1 and p-Smad3 were also detected using a Western blotting assay (n = 7–8). * *p* < 0.05 and ** *p* < 0.01. Data represent the mean ± SD of at least three separate experiments.

**Figure 6 molecules-27-07868-f006:**
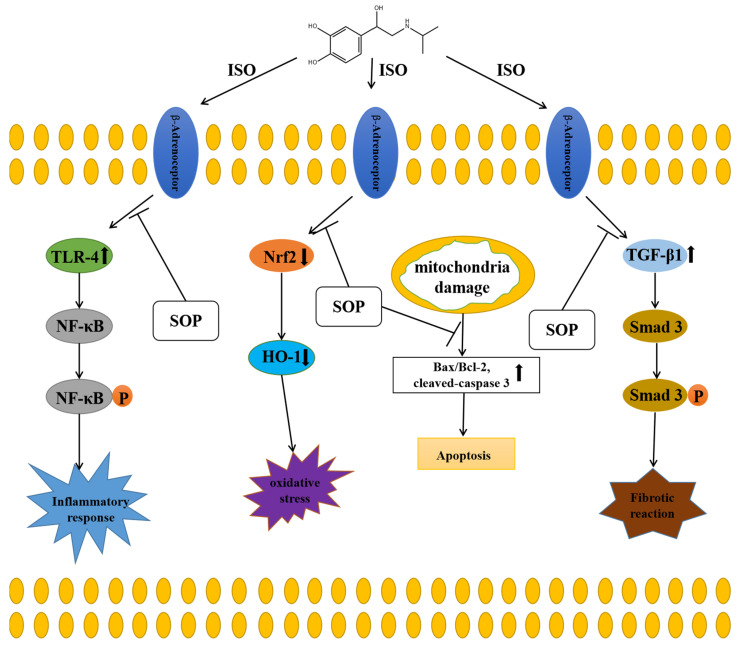
Graphical abstract of the effects of treatment with SOP on ISO-induced kidney injury: SOP could alleviate ISO-induced kidney injury by inhibiting inflammation, apoptosis, oxidative stress and fibrosis; the molecular mechanisms were suppression of the TLR-4/NF-κB and TGF-β1/Smad3 signaling pathways and activation of the Nrf2/HO-1 signaling pathway.

## Data Availability

The data presented in this study are available on request from the corresponding author.

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
