# Peer review of "Sophocarpine Alleviates Isoproterenol-Induced Kidney Injury by Suppressing Inflammation, Apoptosis, Oxidative Stress and Fibrosis"

_molecules, 2022, doi:10.3390/molecules27227868_

Round 1

Reviewer 1 Report

Kidney injuries are prevalent and can be chronic/irreversible. There are a few therapeutic approaches present currently. The author aims to identify, and develop additional effective therapeutic approaches to alleviate kidney injury. The authors created a kidney injury model using ISO and then used SOP as a treatment strategy. SOP is extracted from Sophia flavescens with multiple beneficial properties. The authors studied the impact of SOP on ISO induced kidney injury in mouse models. The authors concluded that SOP could alleviate ISO-induced kidney 26 injury by inhibiting inflammation, apoptosis, oxidative stress and fibrosis. 

Comments to the authors.

ISO is primarily for cardiac dysfunction (pp.52). What is the effect of SOP treatment on cardiac dysfunction in the mice? There should be some comment at the very least about the same in the results and discussion section. What were the observations? 

How is the murine lupus nephritis different from ISO induced kidney dysfunction model? (Pubmed: 30047025) (Line 53-54) would contradict the line in the manuscript. Some of the markers studied in this citation are the similar to the ones studied in the manuscript. What are the similarities and differences between the results post SOP treatment?

All statistical significant comparisons of the SOP treatments are made to the ISO controls. In some situations though such as IL1-beta, pNFKbeta or TNFalpha, the expression of such markers look less than the controls. Is 40mg/kg toxic? Fig 2E. Remarks on such comparisons should be made for all figures and the results should also include if the expression changes to the basal level or not. This should then be extended to include the interpretation of such results.

Discussion on other compounds such as betaine , which have similar therapeutic effects such as SOP on kidney injuries (PubMed: 30811871), should be written in the discussion.

Figure 1 labeling is not easily readable. Taking PBS as an example, is the drug given every day for 2 weeks or given only once after 2 weeks?

Abstract - should be rewritten to include the justification of the study. 

Minor - grammatical and spelling errors

Author Response

Responses for the reviewers comments

Dear editors and Reviewers:

Thank you very much for your decision letter sent to us. We were pleased to have the opportunity to revise our manuscript, and greatly appreciated your helpful comments. We agree to the raised points, and all of the comments have been fully taken into account in this revised version of the manuscript.

We have responded to your comments point by point in the following pages and have revised the manuscript accordingly. We have also attached a copy of the revised manuscript and figures.

All authors have read and agreed on the final version of the manuscript. This manuscript has not been published in whole or in part, and it is not being considered for publication elsewhere.

We are grateful to you and the editorial staff for the review and handling of our manuscript. Likewise, we appreciate your time spent reviewing our manuscript.

Yours sincerely.

Reviewer #1:

(1) Comments and Suggestions for Authors

Kidney injuries are prevalent and can be chronic/irreversible. There are a few therapeutic approaches present currently. The author aims to identify, and develop additional effective therapeutic approaches to alleviate kidney injury. The authors created a kidney injury model using ISO and then used SOP as a treatment strategy. SOP is extracted from Sophia flavescens with multiple beneficial properties. The authors studied the impact of SOP on ISO induced kidney injury in mouse models. The authors concluded that SOP could alleviate ISO-induced kidney 26 injury by inhibiting inflammation, apoptosis, oxidative stress and fibrosis.

Response: Thank you very much for your evaluation of our research. We will revise the manuscript according to your professional advices carefully and response to your comments point by point as follows.

(2) ISO is primarily for cardiac dysfunction (pp.52). What is the effect of SOP treatment on cardiac dysfunction in the mice? There should be some comment at the very least about the same in the results and discussion section. What were the observations? 

Response: We thank you very much and fully agree with your professional perspective on this point. After carefully consideration, as the reviewer said, ISO is usually used to induce cardiac dysfunction. In the present study, the serum cardiac injury biomarkers (CK-MB and LDH) were detected. We found that ISO indeed induced cardiac injury, as evidenced by the up-regulation of CK-MB and LDH challenged by ISO. While this effect was reversed by treatment of SOP (Figure 1D and E). And we added this result to the result and the discussion section, which was marked in red color (We also found that treatment of SOP could alleviate ISO-induced cardiac injury, as evidenced by the down-regulation of serum levels of cardiac injury biomarkers (CK-MB and LDH) when the mice were treated with SOP ). Thank you very much again for your specialized comments on this point.

(3) How is the murine lupus nephritis different from ISO induced kidney dysfunction model ? (Pubmed: 30047025) (Line 53-54) would contradict the line in the manuscript. Some of the markers studied in this citation are the similar to the ones studied in the manuscript. What are the similarities and differences between the results post SOP treatment ?

Response: We thank you very much for your professional comments on this point. The animal model of murine lupus nephritis in the previous research is different from the animal model in our study. ISO is a β Receptor agonist that can be applied to clinical treatment of some related diseases. While ISO also have some corresponding adverse side effects on the body when the drug is used improperly. In the process of scientific research, ISO is usually used for inducing cardiac dysfunction and some related renal diseases. Although these two animal models of renal diseases are different, treatment of SOP has protective effects on both disease models. We disscussed it in our manuscript and this section was marked in red color (Besides, Li, et al. discovered that sophocarpine also alleviate murine lupus nephritis by suppressing NLRP3 inflammasome and NF-κB activation). Thank you very much again for your specialized comments on this point.

(4) All statistical significant comparisons of the SOP treatments are made to the ISO controls. In some situations though such as IL1-beta, pNFKbeta or TNFalpha, the expression of such markers look less than the controls. Is 40mg/kg toxic? Fig 2E. Remarks on such comparisons should be made for all figures and the results should also include if the expression changes to the basal level or not. This should then be extended to include the interpretation of such results.

Response: We thank you very much and fully agree with your specialized advice on this point. Actually we were not careful enough in making Figures and chose the inappropriate protein bands. After reminded by the reviewer, we re-selected some appropriate protein bands to replaced those inappropriate ones in all Figures. Thank you very much again for your specialized comments on this point.

(5) Discussion on other compounds such as betaine , which have similar therapeutic effects such as SOP on kidney injuries (PubMed: 30811871), should be written in the discussion.

Response: We thank you very much and fully agree with your professional perspective on this point. As the reviewer said, betaine has some similar therapeutic effects to SOP. And we discussed it in the discussion section, which was marked in red color (Betaine, also a natural product, was found a protective impact on the renal function in ISO-induced myocardial infarction (MI)). Thank you very much again for your specialized comments on this point.      

(6)Figure 1 labeling is not easily readable. Taking PBS as an example, is the drug given every day for 2 weeksor given only once after 2 weeks?

Response: We thank you very much and fully agree with your specialized advice on this point. Actually, these reagents were given to the mice every day for 2 weeks. And we revised this point in Figure 1A. Thank you very much again for your specialized comments on this point.

(7) Abstract - should be rewritten to include the justification of the study.

Response: We thank you very much for your professional comments on this point. After careful consideration, we added some research backgrounds to the Abstract section to make it more better, which was marked in red color (One of the most common diseases affecting people and leading to high morbidity is kidney injury. Alleviation of inflammation and apoptosis is considered a potential therapeutic approach for kidney injury. Sophocarpine(SOP), a tetracyclic quinolizidine alkaloid, exhibits various beneficial biological properties. ). Thank you very much again for your specialized comments on this point.   

(8) Minor - grammatical and spelling errors.

Response: We thank you very much and fully agree with your professional comments on this point. We have carefully gone through the manuscript and indeed sought out numerous grammatical errors and typos in our manuscript. Therefore, a professional English editor was found to proof the manuscript and reduce the grammatical errors and typos of our manuscript. Thank you very much again for your specialized comments on this point.

Reviewer 2 Report

   In the presented work, the authors used mice model to investigate the effects of sophocarpine (SOP) on isoproterenol (ISO)-induced kidney injury. The authors presented evidences that kidney injury serum biomarkers, SCr and BUN, increased after ISO challenge, while this effect was reversed by treatment of SOP. Pathological changes induced by ISO were also reversed by treatment of SOP in stainings. Inflammatory cytokines, IL-β, IL-6, TNF-α, MCP-1 and NLRP3, increased after challenged with ISO, while decreased by treatment of SOP. Apoptotic proteins cleaved-caspase-3 and Bax increased while Bcl-2 decreased after challenged with ISO, these effects were reversed by treatment of SOP. Antioxidant proteins, SOD-1 and SOD-2, decreased after stimulated by ISO, while increased by treatment of SOP. Fibrotic proteins, Collagen â… , Collagen â…¢, α-SMA, Fibronectin, MMP-2 and MMP-9, increased after challenged by ISO, while decreased by treatment of SOP. Furthermore, the authors discovered that TLR-4/NF-κB and TGF-β1/Smad3 signaling pathways were suppressed while Nrf2/HO-1 signaling pathway was activated. In the presented work, the authors found SOP could alleviate ISO-induced kidney injury by inhibiting inflammation, apoptosis, oxidative stress and fibrosis. The molecular mechanisms were suppression of TLR-4/NF-κB and TGF-β1/Smad3 signaling pathways while activation of Nrf2/HO-1 signaling pathway, indicating that SOP might serve as a novel therapeutic strategy for kidney injury. 

  This is an interesting work, and should be interested by the readers in the field. 

I have some comments.

Line 34-36: reference 1 is not very suitable for this statement. It looks like they are talking about different thing.

Line 36-37: it is better to cite a review paper, not research article. And reference 2 looks not very suitable for this statement.

Line 45-46: reference 5 is an Inappropriate reference. Reference only talked about the electrophysiological mechanisms of sophocarpine. It should not support completely this statement.

Line 46-48: reference 6 never talks about anti-inflammatory, anti-tumor, anti-nociceptive, neuroprotective and immune regulatory properties of sophocarpine.

Line51-53: reference 9 is not very well to support this statement. In this sentence, the authors mention ISO is well-known for..., reference 9 is only case and also this is a case from rat. It is better to replace reference 9 with a review paper that support "well-known.

Line 82-83: the authors should state how the mice were sacrificed.

Figure 1-5: the authors didn't show how the effect of SOP alone.

Figure 6: the authors should explain figure 6 in the text. But in the presented manuscript, the authors only mention it in Line 318, have not explained it in the text or legend.

Author Response

Responses for the reviewers comments

Dear editors and Reviewers:

Thank you very much for your decision letter sent to us. We were pleased to have the opportunity to revise our manuscript, and greatly appreciated your helpful comments. We agree to the raised points, and all of the comments have been fully taken into account in this revised version of the manuscript.

We have responded to your comments point by point in the following pages and have revised the manuscript accordingly. We have also attached a copy of the revised manuscript and figures.

All authors have read and agreed on the final version of the manuscript. This manuscript has not been published in whole or in part, and it is not being considered for publication elsewhere.

We are grateful to you and the editorial staff for the review and handling of our manuscript. Likewise, we appreciate your time spent reviewing our manuscript.

Yours sincerely.

Reviewer #2:

(1) Comments and Suggestions for Authors

In the presented work, the authors used mice model to investigate the effects of sophocarpine (SOP) on isoproterenol (ISO)-induced kidney injury. The authors presented evidences that kidney injury serum biomarkers, SCr and BUN, increased after ISO challenge, while this effect was reversed by treatment of SOP. Pathological changes induced by ISO were also reversed by treatment of SOP in stainings. Inflammatory cytokines, IL-β, IL-6, TNF-α, MCP-1 and NLRP3, increased after challenged with ISO, while decreased by treatment of SOP. Apoptotic proteins cleaved-caspase-3 and Bax increased while Bcl-2 decreased after challenged with ISO, these effects were reversed by treatment of SOP. Antioxidant proteins, SOD-1 and SOD-2, decreased after stimulated by ISO, while increased by treatment of SOP. Fibrotic proteins, Collagen â… , Collagen â…¢, α-SMA, Fibronectin, MMP-2 and MMP-9, increased after challenged by ISO, while decreased by treatment of SOP. Furthermore, the authors discovered that TLR-4/NF-κB and TGF-β1/Smad3 signaling pathways were suppressed while Nrf2/HO-1 signaling pathway was activated. In the presented work, the authors found SOP could alleviate ISO-induced kidney injury by inhibiting inflammation, apoptosis, oxidative stress and fibrosis. The molecular mechanisms were suppression of TLR-4/NF-κB and TGF-β1/Smad3 signaling pathways while activation of Nrf2/HO-1 signaling pathway, indicating that SOP might serve as a novel therapeutic strategy for kidney injury. 

This is an interesting work, and should be interested by the readers in the field. 

Response: Thank you very much for your evaluation of our research. We will revise the manuscript according to your professional advices carefully and response to your comments point by point as follows.

(2) Line 34-36: reference 1 is not very suitable for this statement. It looks like they are talking about different thing.

Response: We thank you very much and fully agree with your professional comments on this point. After carefully searched references, we replaced an appropriate one ([1] Lei L, Li L, Zhang H. Advances in the Diagnosis and Treatment of Acute Kidney Injury in Cirrhosis Patients. Biomed Res Int. 2017;2017:8523649.). Thank you very much again for your specialized comments on this point.  

(3) Line 36-37: it is better to cite a review paper, not research article. And reference 2 looks not very suitable for this statement.

Response: We thank you very much and fully agree with your professional comments on this point. After carefully searched references, we replaced an appropriate one ([2] Patel M, Gbadegesin RA. Update on prognosis driven classification of pediatric AKI. Front Pediatr. 2022;10:1039024.). Thank you very much again for your specialized comments on this point.

(4) Line 45-46: reference 5 is an Inappropriate reference. Reference only talked about the electrophysiological mechanisms of sophocarpine. It should not support completely this statement.

Response: We thank you very much and fully agree with your professional comments on this point. After carefully searched references, we replaced an appropriate one ([5] Li Y, Wang G, Liu J, Ouyang L. Quinolizidine alkaloids derivatives from Sophora alopecuroides Linn: Bioactivities, structure-activity relationships and preliminary molecular mechanisms. Eur J Med Chem. 2020 Feb 15;188:111972.). Thank you very much again for your specialized comments on this point.

(5) Line 46-48: reference 6 never talks about anti-inflammatory, anti-tumor, anti-nociceptive, neuroprotective and immune regulatory properties of sophocarpine.

Response: We thank you very much and fully agree with your professional comments on this point. After carefully searched references, we replaced appropriate ones ([6] Wang FL, Wang H, Wang JH, Wang DX, Gao Y, Yang B, Yang HJ, Ji YB, Xin GS. Analgesic and Anti-Inflammatory Activities of Sophocarpine from Sophora viciifolia Hance. Biomed Res Int. 2021;2021:8893563. [7] Wang Q, Wang T, Zhu L, He N, Duan C, Deng W, Zhang H, Zhang X. Sophocarpine Inhibits Tumorgenesis of Colorectal Cancer via Downregulation of MEK/ERK/VEGF Pathway. Biol Pharm Bull. 2019 Nov 1;42(11):1830-1838. [8] Zhu X, Gu Z, Yu Y, Yang W, Li M, Li Y, Zhang P, Wang J, Zhao J. Neuronal Apoptosis Preventive Potential of Sophocarpine via Suppression of Aβ-Accumulation and Down-Regulation of Inflammatory Response. Dokl Biochem Biophys. 2021 Mar;497(1):116-122. [9] Li C, Gao Y, Tian J, Shen J, Xing Y, Liu Z. Sophocarpine administration preserves myocardial function from ischemia-reperfusion in rats via NF-κB inactivation. J Ethnopharmacol. 2011 Jun 1;135(3):620-5.). Thank you very much again for your specialized comments on this point.

(6) Line51-53: reference 9 is not very well to support this statement. In this sentence, the authors mention ISO is well-known for..., reference 9 is only case and also this is a case from rat. It is better to replace reference 9 with a review paper that support "well-known.

Response: We thank you very much and fully agree with your professional comments on this point. After carefully searched references, we replaced an appropriate one ([13] Allawadhi P, Khurana A, Sayed N, Kumari P, Godugu C. Isoproterenol-induced cardiac ischemia and fibrosis: Plant-based approaches for intervention. Phytother Res. 2018 Oct;32(10):1908-1932.). Thank you very much again for your specialized comments on this point.

(7) Line 82-83: the authors should state how the mice were sacrificed.

Response: We thank you very much for your professional comments on this point. After careful consideration, we added this portion to the Materials and Methods section to make it more better, which was marked in red color (After anesthetization by pentobarbital sodium (30mg/kg) intraperitoneally, the vacuum tubes were used to collect blood from the right ventricle of the mice, then the mice were killed for further study.). Thank you very much again for your specialized comments on this point. 

(8) Figure 1-5: the authors didn't show how the effect of SOP alone.

Response: We thank you very much for your professional comments on this point. As the reviewer said, it is important to detect the effect of SOP in mice alone. While we did not done relevant studies in this section. It is a big limitation for our manuscript and we added it to the limitations section, which was marked in red color (the toxicity of SOP in mice was not be evaluated in the present study. ). And we will further investigate in this part in our further study. Thank you very much again for your specialized comments on this point.

(9) Figure 6: the authors should explain figure 6 in the text. But in the presented manuscript, the authors only mention it in Line 318, have not explained it in the text or legend.

Response: We thank you very much for your professional comments on this point. As the reviewer said, there was no explanation for Figure 6. After careful consideration, we added it to the Figure 6, which was marked in red color (SOP could alleviate ISO-induced kidney injury by inhibiting inflammation, apoptosis, oxidative stress and fibrosis; The molecular mechanisms were suppression of TLR-4/NF-κB and TGF-β1/Smad3 signaling pathways while activation of Nrf2/HO-1 signaling pathway. ).

Reviewer 3 Report

Zhou et. al investigated the protective effect of sophocarpine (SOP) on soproterenol (ISO) -induced kidney damage. Sophocarpine, extracted from Sophora flavescens, has proven anti-inflammatory, anti-tumor, anti-nociceptive, neuroprotective and immune regulatory properties. Therefore, it seems justified to investigate whether SOP may have a protective effect on the kidneys exposed to toxic substances.

The wide range of research methods used in the experiment deserves great recognition. The authors of the study used an animal model - the control and study groups are sufficiently numerous. HE, Mason and TUNEL staining, ELISA and spectrophotometric tests and Western-Blott were used to determine the SOP properties. I think that the methodology should be more widely described - it is worth presenting the principle of the test on which it was worked (especially in, Detection of serum malondialdehyde (MDA), superoxide dismutase (SOD) and glutathione (GSH) levels). The figures presented by the authors are very legible and facilitate the understanding of the text. I believe that the posted microscopic images should be larger and with better resolution. The graphs present the reported results well, but again, they should be larger. I believe that it is possible to expand the discussion and confront the obtained results with other reports.

Nevertheless, I congratulate the Authors on their work and recommend the article for publication after minor corrections.

Author Response

Responses for the reviewers comments

Dear editors and Reviewers:

Thank you very much for your decision letter sent to us. We were pleased to have the opportunity to revise our manuscript, and greatly appreciated your helpful comments. We agree to the raised points, and all of the comments have been fully taken into account in this revised version of the manuscript.

We have responded to your comments point by point in the following pages and have revised the manuscript accordingly. We have also attached a copy of the revised manuscript and figures.

All authors have read and agreed on the final version of the manuscript. This manuscript has not been published in whole or in part, and it is not being considered for publication elsewhere.

We are grateful to you and the editorial staff for the review and handling of our manuscript. Likewise, we appreciate your time spent reviewing our manuscript.

Yours sincerely.

Reviewer #3:

Comments and Suggestions for Authors

Zhou et. al investigated the protective effect of sophocarpine (SOP) on soproterenol (ISO) -induced kidney damage. Sophocarpine, extracted from Sophora flavescens, has proven anti-inflammatory, anti-tumor, anti-nociceptive, neuroprotective and immune regulatory properties. Therefore, it seems justified to investigate whether SOP may have a protective effect on the kidneys exposed to toxic substances.

The wide range of research methods used in the experiment deserves great recognition. The authors of the study used an animal model - the control and study groups are sufficiently numerous. HE, Mason and TUNEL staining, ELISA and spectrophotometric tests and Western-Blott were used to determine the SOP properties. I think that the methodology should be more widely described - it is worth presenting the principle of the test on which it was worked (especially in, Detection of serum malondialdehyde (MDA), superoxide dismutase (SOD) and glutathione (GSH) levels). The figures presented by the authors are very legible and facilitate the understanding of the text. I believe that the posted microscopic images should be larger and with better resolution. The graphs present the reported results well, but again, they should be larger. I believe that it is possible to expand the discussion and confront the obtained results with other reports.

Nevertheless, I congratulate the Authors on their work and recommend the article for publication after minor corrections.

Response: Thank you very much for your evaluation of our research. Firstly, we added detailed descriptions of the methods in the Materials and Methods section, like Detection of serum malondialdehyde (MDA), superoxide dismutase (SOD) and glutathione (GSH) levels, and the revised portions were marked in red color. Besides, the graphs in the Figures were revised significantly for a better view. Thank you very much again for your evaluation of our study.

Round 2

Reviewer 1 Report

The authors have addressed all the comments.